# Cyclin-Dependent Kinase Inhibitor BMI-1026 Induces Apoptosis by Downregulating Mcl-1 (L) and c-FLIP (L) and Inactivating p-Akt in Human Renal Carcinoma Cells

**DOI:** 10.3390/ijms22084268

**Published:** 2021-04-20

**Authors:** Dong Eun Kim, Jinho Lee, Jong Wook Park, Hyunsu Kang, Yu Ri Nam, Taeg Kyu Kwon, Ki-Suk Kim, Shin Kim

**Affiliations:** 1Department of Immunology, School of Medicine, Keimyung University, 1095 Dalgubeol-daero, Dalseo-gu, Daegu 42601, Korea; ehddms9762@daum.net (D.E.K.); j303nih@dsmc.or.kr (J.W.P.); kwontk@dsmc.or.kr (T.K.K.); 2Department of Chemistry, Keimyung University, 1095 Dalgubeol-daero, Dalseo-gu, Daegu 42601, Korea; jinho@kmu.ac.kr; 3R&D Center for Advanced Pharmaceuticals & Evaluation, Korea Institute of Toxicology, Daejeon 34114, Korea; hyunsu.kang@kitox.re.kr (H.K.); yuri.nam@kitox.re.kr (Y.R.N.); 4Institute of Medical Science & Institute of Cancer Research, Keimyung University, 1095 Dalguceol-daero, Dalseo-gu, Daegu 42601, Korea

**Keywords:** BMI-1026, apoptosis, Mcl-1, cellular FADD-like IL-1β-converting enzyme inhibitory protein, p-Akt

## Abstract

Previous studies have investigated the inhibitory effect of BMI-1026 on cyclin-dependent kinase 1 in vitro. However, the molecular mechanisms by which BMI-1026 treatment leads to cancer cell death remain unclear. This study was conducted to investigate the anticancer mechanisms of BMI-1026 on human renal carcinoma Caki cells. BMI-1026 induced apoptosis in association with the cleavage of poly(ADP-ribose) polymerase and pro-caspase-3 and the release of apoptosis-inducing factor and cytochrome *c* from mitochondria in Caki cells. BMI-1026-induced apoptosis was inhibited by the pan-caspase inhibitor z-VAD-fmk. Furthermore, BMI-1026 downregulated Bcl-2 and X-linked inhibitor of apoptosis protein (XIAP) at the transcriptional level and Mcl-1 (L) and cellular FADD-like IL-1β-converting enzyme inhibitory protein (c-FLIP (L)) at the post-transcriptional level. Interestingly, Mcl-1 (L) and c-FLIP (L), but not Bcl-2 or XIAP, played important roles in BMI-1026-induced Caki cell apoptosis. Although the constitutively active form of Akt did not attenuate BMI-1026-induced apoptosis, blockade of the PI3K/Akt pathway using a subcytotoxic concentration of the PI3K/Akt inhibitor LY294002 enhanced Caki cell apoptosis induced by BMI-1026. Electrophysiological safety was confirmed by determining the cardiotoxicity of BMI-1026 via left ventricular pressure analysis. These results suggest that BMI-1026 is a potent multitarget anticancer agent with electrophysiological safety and should be further investigated.

## 1. Introduction

The human genome encodes 538 protein kinases [1], and the identification of their roles in cancer has led to extensive studies on the development and potential uses of protein kinase inhibitors for anticancer therapy [2]. Among the various targets of protein kinase inhibitors, cyclin-dependent kinases (Cdks) are promising targets for cancer therapeutics because of their important role in cancer cell proliferation via cell cycle dysregulation, which is a hallmark of cancer [3]. To develop potent anticancer agents that inhibit Cdk activity, a large amount of research has focused on screening for ATP-competitive inhibitors and investigating the mechanisms by which they exert their anticancer effects [3].

BMI-1026, a synthetic 2-aminopyrimidine analogue, is a potential anticancer agent that targets Cdks [4]. Recent studies demonstrated that BMI-1026 has anticancer effects, such as mitotic catastrophe-induced cancer cell death and the induction of premature senescence in cancer cells [4,5]. Moreover, BMI-1026 can inhibit phosphorylation of epithelial cell transforming 2 in G2/M phase [6] in addition to inducing germinal vesicle breakdown of immature oocytes [7]. Additionally, BMI-1026 causes cytokinesis via Cdk1 inhibition [8] and activates metaphase II-arrested oocytes [7]. However, previous studies of BMI-1026 have not investigated its regulatory effects on apoptosis-related genes and their related signaling mechanisms.

The success rate of drug development is determined by efficacy and safety. Drug-induced cardiotoxicity is directly related to patient lifespan, which is a major factor leading to the discontinuation of development or removal of commercial drugs from the market. According to previous studies, it is important to ensure the electrophysiological safety of drugs based on reliable assessment of the potential for proarrhythmic cardiotoxicity. Such methods include QT interval prolongation, which increases the risk of torsade de pointes (TdP) [9,10,11]. TdP prolongs the QT interval of the electrocardiogram by inhibiting the human ether-à-go-go-related gene (hERG) potassium channel, which is related to the action potential and its duration in the myocardial cell membrane [12,13]. Recommended experiments for assessing cardiotoxicity include assessing blood pressure and heart rate, electrocardiograms, and ventricular contraction studies in vivo [10,11].

In the present study, we demonstrated the molecular mechanism of the anticancer effect of BMI-1026 and its potential as an anticancer agent for the first time. We also confirmed the electrophysiological safety of BMI-1026 by examining its effects via left ventricular pressure (LVP) analysis. Moreover, we showed that BMI-1026 induced Caki cell apoptosis by downregulating the expression of anti-apoptotic proteins Mcl-1 (L) and cellular FADD-like IL-1β-converting enzyme inhibitory protein (c-FLIP (L)).

## 2. Results

### 2.1. BMI-1026 Has Antiproliferative Effects and Induces Apoptosis in Various Human Cancer Cells

To investigate the anticancer effect of BMI-1026, Caki cells were treated with various concentrations of BMI-1026 for the indicated time periods. Using the automated live cell analyzer Incucyte^®^, we assayed the inhibitory effect of BMI-1026 on Caki cell growth. As shown in Figure 1b, BMI-1026 inhibited Caki cell growth in a dose-dependent manner. Moreover, as determined via light microscopy, treatment with BMI-1026 exhibited progressive morphological changes typical of apoptosis, such as cell shrinkage, apoptotic body formation, and cell detachment from the plate (Figure 1c). To evaluate the effect of BMI-1026 on cell cycle distribution, the sub-G1, G0/G1, S, and G2/M populations were analyzed using flow cytometry. BMI-1026 induced G2/M arrest and apoptosis in a dose- and time-dependent manner (Figure 1d,e). Next, a clonogenic assay was performed to assess the effectiveness of BMI-1026 on the reproductive potential of Caki cells. As shown in Figure 1f, the clonogenic survival of Caki cells was decreased by treatment with nanomolar concentrations of BMI-1026. Additionally, BMI-1026 impaired the migration of Caki cells compared to in the control (Figure 1g). We next investigated whether BMI-1026 induces cell cycle arrest and apoptosis in several types of human cancer cells. As shown in Figure 1h, treating DU145, U937, and HCT116 cells with BMI-1026 resulted in marked dose-dependent increases in the accumulation of the G2/M arrest and the sub-G1 population. Taken together, these results suggest that BMI-1026 potently induces G2/M arrest and apoptosis in various human cancer cells, including Caki cells.

### 2.2. BMI-1026 Regulates Apoptosis-Related Proteins in Caki Cells

Next, we evaluated the effect of the pan-caspase inhibitor benzyloxy carbony-Val-Ala-Asp-fluoromethyl ketone (z-VAD-fmk) on BMI-1026-induced apoptosis in Caki cells. BMI-1026 increased the sub-G1 population of Caki cells and induced poly(ADP-ribose) polymerase (PARP) and caspase-3 cleavage, which were remarkably suppressed by pre-treatment with z-VAD-fmk (Figure 2a). Mitochondria play a critical role in apoptosis by releasing AIF and cytochrome *c* [14,15]. As shown in Figure 2b, BMI-1026 induced the dose-dependent release of AIF and cytochrome *c* into the cytoplasm in Caki cells. To identify the underlying mechanisms involved in BMI-1026-induced apoptosis, we analyzed the expression levels of apoptosis-related proteins in BMI-1026-treated Caki cells. Treatment of the cells with BMI-1026 resulted in an increase in PARP cleavage and dose- and time-dependent downregulation of XIAP, c-FLIP (L), Bcl-2, and Mcl-1 (L) (Figure 2c,d). z-VAD-fmk pre-treatment did not restore the levels of downregulated XIAP, c-FLIP (L), Bcl-2, and Mcl-1 (L) (Figure 2e). To further investigate whether XIAP, c-FLIP (L), Bcl-2, and Mcl-1 (L) downregulation is mediated at the transcriptional level, the mRNA expression levels of the genes were evaluated by real-time polymerase chain reaction (PCR) and reverse transcription PCR. As shown in Figure 2f, BMI-1026 decreased the mRNA levels of XIAP and Bcl-2, whereas those of c-FLIP (L) and Mcl-1 (L) did not change. Next, we examined the stability of c-FLIP (L) and Mcl-1 (L) proteins after treatment with BMI-1026. The c-FLIP (L) and Mc-1 (L) protein levels rapidly decreased in the presence of cycloheximide and were significantly lower in BMI-1026-treated cells than in vehicle-treated cells (Figure 2g). Next, we investigated Mcl-1 (L) and c-FLIP (L) regulation by BMI-1026 at the post-translational level. As shown in Figure 2h, proteasome activity was partially involved in the downregulation of Mcl-1 (L) and c-FLIP (L) expression in BMI-1026-treated cells.

### 2.3. BMI-1026-Induced Apoptosis Is Associated with Various Apoptosis-Related Proteins in Caki Cells

To investigate the role of apoptosis-related proteins in BMI-1026-induced apoptosis, we used human renal carcinoma Caki cells engineered to overexpress Bcl-2 (Caki/Bcl-2), c-FLIP (L) (Caki/c-FLIP (L)), and Mcl-1 (L) (Caki/Mcl-1 (L)). As shown in Figure 3a, c-FLIP (L) overexpression markedly suppressed BMI-1026-induced apoptosis. Mcl-1 (L) overexpression partially attenuated BMI-1026-induced apoptosis, whereas Bcl-2 overexpression did not inhibit this effect. To evaluate the role of XIAP in BMI-1026-induced apoptosis, we transfected Caki cells with small interfering RNA (siRNA) targeting XIAP mRNA and treated the cells with or without BMI-1026. As shown in Figure 3b, BMI-1026-induced accumulation of the sub-G1 phase was not enhanced in cells transfected with XIAP siRNA compared to in control siRNA-transfected cells. Akt/protein kinase B plays an important role in promoting survival and inhibiting apoptosis [16,17]. To investigate whether Akt proteins have anti-apoptotic potential in BMI-1026-induced apoptosis, we used human renal carcinoma Caki cells engineered to express the constitutively active form of Akt (Caki/Akt-DA). As shown in Figure 3c,d, BMI-1026 treatment led to increased PARP cleavage and sub-G1 phase cells in both the Caki/vector and Caki/Akt-DA groups. Moreover, combined treatment with subcytotoxic concentrations of BMI-1026 and LY294002 induced apoptosis (Figure 3e). Apoptosis is regulated by various components, including p53, an important tumor suppressor [18], and p53 upregulated modulator of apoptosis (PUMA), a pro-apoptotic member of the Bcl-2 protein family [19]. To evaluate the functional role of p53 and PUMA in BMI-1026-induced apoptosis, we first identified whether BMI-1026 regulates p53 and PUMA. As shown in Figure 4a,b, BMI-1026 upregulated p53 and PUMA in a dose- and time-dependent manner. Next, we transfected Caki cells with siRNA targeting mRNAs of p53 and PUMA, then treated the cells with or without BMI-1026. Immunoblot analysis demonstrated that transfection with siRNA of p53 and PUMA resulted in suppression of p53 and PUMA expression in Caki cells compared to in cells transfected with control green fluorescent protein siRNA (Figure 4c,d). Interestingly, BMI-1026-induced accumulation of cells in sub-G1 phase was not altered in cells transfected with siRNA of p53 and PUMA compared to in control siRNA-transfected cells (Figure 4c,d). Collectively, these results suggest that Mcl-1 (L) and c-FLIP (L) protein downregulation has functional significance in BMI-1026-induced Caki cell apoptosis.

### 2.4. BMI-1026-Induced Apoptosis Is Independent of Calpain, MAPK, and ROS Signaling Pathways

To explore the signaling pathway involved in BMI-1026-induced apoptosis, we used specific inhibitors. Our results revealed that neither specific calpain inhibitors (calpain III inhibitor, z-Leu-Leu-CHO, and Ac-Leu-Leu-Nle-CHO (ALLN)) nor specific mitogen-activated protein kinase (MAPK) inhibitors (PD, MEK inhibitor; SP, JNK inhibitor; SB, p38 inhibitor) affected BMI-1026-induced apoptosis in Caki cells (Figure 5a,b). Additionally, as shown in Figure 5c, *N*-acetylcysteine and glutathione reduced ethyl ester did not attenuate BMI-1026-induced apoptosis in Caki cells. These data suggest that BMI-1026-induced apoptosis is not associated with calpain, the MAPK pathway, or reactive oxygen species generation.

### 2.5. BMI-1026 Has Electrophysiological Safety

During drug development and efficacy assessment, it is important to perform LVP analysis in vivo to check for drug-induced cardiotoxicity and safety [11]. To determine the cardiotoxic effects of BMI-1026, we conducted LVP analysis to measure the heart rate, left ventricular end systolic pressure, left ventricular developed pressure, left ventricular end-diastolic pressure, and maximum or minimum left ventricular contraction force (dP/dt maximum or minimum). Our results showed that the left ventricular end-systolic pressure and left ventricular developed pressure were decreased at 10 and 100 nM BMI-2016 (Figure 6a,b). The dP/dt minimum is a useful index of contractility, which is the rate of change of the ventricular pressure pulse. The dP/dt was decreased at only 100 nM BMI-1036 (Figure 6f). However, the left ventricular end-diastolic pressure, heart rate, and dP/dt maximum had no effect on BMI-1026 (Figure 6c–e). These results suggest that BMI-1026 has little effect on the heart and demonstrate that BMI-1026 has electrophysiological safety.

## 3. Discussion

Exploring major anticancer targets and their underlying mechanisms has contributed to the development of specific and targeted chemotherapeutic agents [20]. However, intratumoral heterogeneity is a major obstacle to chemotherapeutic cancer treatment [21]. Overcoming tumor heterogeneity by developing multitarget anticancer agents may therefore be an improved strategy for chemotherapy. BMI-1026 is an anticancer drug developed as a Cdk1 inhibitor and has anticancer effects [4,5]. In the present study, we found that BMI-1026 effectively induced cell cycle arrest and apoptotic cell death in cancer cells. Caspase-3 activation was associated with BMI-1026-induced apoptosis. Remarkably, BMI-1026-induced apoptosis depended on Mcl-1 (L) and c-FLIP (L) downregulation. Our results suggest the potential for using BMI-1026 to act as a multitarget anticancer agent.

Apoptosis, or type I programmed cell death, is characterized by the following morphological features: blebbing, cell shrinkage, chromatin condensation, chromosomal DNA fragmentation, and nuclear fragmentation [22]. Under microscopic evaluation, BMI-1026 induced apoptotic morphological changes and inhibited migration in Caki cells (Figure 1c,h). Activation of caspase-3, a factor for cell death-inducing proteases, leads to PARP cleavage [23]. We found that BMI-1026 induces caspase-3 activation, which subsequently cleaves PARP (Figure 2a). Mitochondria are intracellular organelles that play an important role in apoptosis by releasing AIF and cytochrome *c*, leading to activation of the caspase-3 pathway [24]. As shown in Figure 2b, BMI-1026 treatment for 24 h resulted in the release of AIF and cytochrome *c* from the mitochondria to the cytosol. Additionally, PARP cleavage was induced by BMI-1026-mediated caspase-3 activation after 12 h of treatment. These data suggest that AIF and cytochrome *c* were released from mitochondria prior to 12 h of BMI-1026 treatment, leading to caspase-3 activation and PARP cleavage.

Apoptosis is largely controlled by anti-apoptotic family proteins such as Bcl-2 [25], Mcl-1 (L) [26], c-FLIP (L) [27], and XIAP [28] and pro-apoptotic family proteins such as PUMA [29]. Upon DNA damage, the tumor suppressor p53 plays an important role in apoptosis [30]. In this study, we evaluated the effect of BMI-1026 on the apoptosis-related proteins in Caki cells. We found that BMI-1026 upregulated p53 and PUMA (Figure 4a,b) and downregulated anti-apoptotic proteins including Bcl-2, Mcl-1 (L), XIAP, and c-FLIP (L) (Figure 2c,d). Interestingly, Bcl-2 and XIAP are downregulated by BMI-1026 at the transcriptional level, whereas BMI-1026 induced proteasome-dependent Mcl-1 (L) and c-FLIP (L) downregulation at the post-transcriptional level (Figure 2f,g). Moreover, Bcl-2 and XIAP overexpression in Caki cells inhibited BMI-1026-induced apoptosis (Figure 3a). However, neither p53 siRNA nor PUMA siRNA blocked BMI-1026-induced apoptosis (Figure 4c). Our data suggest that Bcl-2 and XIAP play important roles in BMI-1026-induced Caki cell apoptosis.

Activation of the PI3K/Akt pathway reportedly plays an important role in promoting cell survival [31], suggesting that it is a promising candidate as a chemotherapeutic anticancer target. Additionally, the well-known Cdk inhibitors flavopiridol and roscovitine and the PI3K inhibitor LY294002 have been shown to synergistically induce human leukemia cell apoptosis [32]. We therefore evaluated whether the PI3K/Akt pathway can be a target of BMI-1026. As shown in Figure 3c, BMI-1026 inactivated (phospho-Akt) p-Akt in a time- and dose-dependent manner. Moreover, we found that BMI-1026 overcame constitutively active Akt, and combined treatment with subcytotoxic concentrations of BMI-1026 and LY294002 induced apoptosis (Figure 3d,e). These results suggest that combined therapy with BMI-1026 and PI3K/Akt inhibition is a promising anticancer strategy for Akt-overexpressing cancer cells.

Cardiovascular toxicity and safety testing is mandatory during drug development. TdP phenomenon, a fatal symptom of heart arrhythmia, was found to inhibit hERG channels in the myocardial membrane, causing QT interval prolongation [10]. According to existing guidelines, all medicines are tested to ensure their electrophysiological safety from cardiotoxicity in vivo and in vitro [9,10,11]. In this study, we confirmed the electrophysiological safety of BMI-1026 by determining its cardiotoxicity via LVP analysis. As a result, we confirmed the safety of BMI-1026 (Figure 6). However, to further ensure its electrophysiological safety, various in vitro and in vitro experiments are required [11]. Our further studies will focus on the cardiovascular safety of BMI-1026 via hERG analysis or electrocardiogram telemetry in vivo. In addition, we will confirm its electrophysiological safety through a comprehensive in vitro proarrhythmia assay. This assay can reduce the clinical ratio and predict clinical outcomes based on in vitro experiments. It is expected that electrophysiological safety can be secured by using in silico models, deriving results similar to those that would be observed in vivo based on in vitro results [33]. Through safety assessments using the peak Na, late Na, Ca, and hERG channels, which are representative ion channels for the QT interval, we expect that electrophysiological safety can be enhanced by applying in vitro data to in silico models.

In this study, we investigated the molecular mechanisms of the anticancer effects of BMI-1026 in Caki cells. BMI-1026-induced apoptosis is dependent on caspase-3 activation, the release of AIF and cytochrome *c*, and Mcl-1 (L) and c-FLIP (L) downregulation. Moreover, we demonstrated the electrophysiological safety of BMI-1026 in vitro and in vivo. Taken together, these results suggest that BMI-1026 is a promising anticancer agent that warrants further clinical evaluation.

## 4. Materials and Methods

### 4.1. Cell Lines and Culture

Caki human kidney cancer cells line, HCT116 human colorectal carcinoma cell line, and DU145 human prostate cancer cells line were obtained from the American Type Culture Collection (ATCC, Manassas, VA, USA) and grown in Dulbecco’s modified Eagle’s medium (DMEM, Welgene Inc., Gyeongsan, Korea) containing 10% heat-inactivated fetal bovine serum (FBS, Welgene Inc., Gyeongsan, Korea), 20 mM HEPES buffer, 100 μg/mL streptomycin, and 100 μg/mL penicillin. U937 human leukemia cancer cells line was obtained from ATCC (Manassas, VA, USA) and grown in RPMI 1640 medium (Welgene Inc., Gyeongsan, Korea) supplemented with 10% heat-inactivated FBS (Welgene Inc., Gyeongsan, Korea), 2 mM L-glutamine, 100 μg/mL streptomycin, and 100 μg/mL penicillin.

### 4.2. Drugs and Materials

BMI-1026 (2,4-bis(2-aminopyrimidin-4-yl)phenol) was synthesized in six steps from commercially available 4′-hydroxyacetophenone. The structure of BMI-1026 was confirmed by ^1^H NMR and ESI-MS: ^1^H NMR (500 MHz, DMSO-*d*_6_): δ 8.73 (s, 1H), 8.47 (d, *J* = 5.2 Hz, 1H), 8.39 (d, *J* = 5.7 Hz, 1H), 8.23 (d, *J* = 8.6 Hz, 1H), 7.86 (s, 2H), 7.58 (d, *J* = 5.2 Hz, 1H), 7.48 (s, 1H), 7.39 (s, 2H), 7.10 (d, *J* = 8.6 Hz, 1H)); ESI-MS: m/z = 281.04 [M + H]^+^. The purity of the compound was >98.6%, as determined by HPLC. Anti-Bcl-2 (sc-783, 1:700), anti-cytochrome *c* (sc-23983, 1:700), anti-Mcl-1 (L) (sc-819, 1:1000), anti-c-FLIP (sc-8347, 1:700), anti-apoptosis-inducing factor (AIF, sc-5586, 1:700), anti-p53 (sc-126, 1:1000), and anti-PUMA (sc-19187, 1:700) antibodies were purchased from Santa Cruz Biotechnology (Dallas, TX, USA). Anti-p-Akt (9271s, 1:700), anti-PARP (9542s, 1:700), and anti-cleaved caspase-3 (9661s, 1:700) antibodies were purchased from Cell Signaling Technology (Danvers, MA, USA). Anti-caspase-3 (610322, 1:700) and anti-XIAP (610762, 1:1000) antibodies were purchased from BD Biosciences Pharmingen (San Jose, CA, USA). Anti-MnSOD antibody (06984, 1:1000) was purchased from Millipore (Billerica, MA, USA). Anti-β-actin antibody (A5441, 1:10,000), glutathione reduced ethyl ester, calpain inhibitor III, and cycloheximide were obtained from Sigma-Aldrich (St. Louis, MO, USA). z-VAD-fmk was purchased from R&D Systems (Minneapolis, MN, USA). PD98059 (MEK inhibitor, PD), SP600125 (JNK inhibitor, SP), SB-203580 (p38 MAP kinase inhibitor, SB), z-Leu-Leu-CHO, ALLN, and LY 294002 were purchased from Enzo Life Sciences (Farmingdale, NY, USA). *N*-Acetylcysteine and MG132 were purchased from Calbiochem (San Diego, CA, USA).

### 4.3. Western Blotting Analysis

Cellular lysates were prepared by suspending 1 × 10^6^ cells in 100 μL lysis buffer (137 mM NaCl, 15 mM ethylene glycol tetraacetic acid, 0.1 mM sodium orthovanadate, 15 mM MgCl_2_, 0.1% Triton X-100, 25 mM 3-(*N*-morpholino)propanesulfonic acid, 100 μM phenylmethylsulfonyl fluoride, and 20 μM leupeptin, adjusted to pH 7.2). The cells were disrupted by sonication and extracted at 4 °C for 30 min. The lysates were centrifuged at 10,000 *g* for 15 min at 4 °C, and the supernatant fractions were collected. Approximately 50 μg of protein was separated by sodium dodecyl sulfate polyacrylamide gel electrophoresis (SDS-PAGE). The proteins were electrotransferred to Immobilon-P membranes (Millipore, Billerica, MA, USA). The membranes were incubated in blocking buffer (0.05% Tween^®^ 20 with 5% non-fat dry milk in TBS) for 3 h. After washing three times with TBST, the membranes were incubated with primary antibody overnight. Detection of specific proteins was carried out with an ECL Western blotting kit (Millipore, Billerica, MA, USA) according to the manufacturer’s instructions. Signal intensity was analyzed using the Chemi Image documentation system (Fusion Rs7, VILBER LOUTMAT, Collégien, France).

### 4.4. Flow Cytometric Analysis

Approximately 0.5 × 10^6^ cells were suspended in 100 μL phosphate-buffered saline (PBS), and 200 μL of 95% ethanol was added while vortexing. The cells were incubated at 4 °C for 1 h, washed with PBS, and resuspended in 250 μL of 1.12% sodium citrate buffer (pH 8.4) together with 12.5 μg RNase. Incubation was continued at 37 °C for 30 min. The cellular DNA was then stained by applying 250 μL propidium iodide (50 μg/mL) for 30 min at room temperature. The stained cells were analyzed by fluorescence-activated cell sorting on a FACScan flow cytometer (Becton Dickinson and Co., Franklin Lakes, NY USA) to determine the relative DNA content based on fluorescence.

### 4.5. Analysis of Mitochondrial AIF and Cytochrome c Release

Approximately 0.3 × 10^6^ Caki cells were harvested, washed once with ice-cold PBS, and gently lysed for 2 min in 80 μL ice-cold lysis buffer (250 mM sucrose, 1 mM EDTA, 20 mM Tris-HCl pH 7.2, 1 mM DTT, 10 mM KCl, 1.5 mM MgCl2, 5 μg/mL pepstatin A, 10 μg/mL leupeptin, 2 μg/mL aprotinin). Lysates were centrifuged at 12,000 *g* at 4 °C for 10 min to obtain the supernatants (cytosolic extracts free of mitochondria) and pellets (fraction that contains mitochondria). Cytosolic protein (30 μg) was resolved by 12% SDS-PAGE, transferred to nitrocellulose, and probed with anti-AIF and anti-cytochrome *c* antibodies.

### 4.6. Reverse Transcription PCR

To obtain cDNA, total RNA was prepared using TRIzol reagent (Life Technologies, Carlsbad, CA, USA) and M-MLV reverse transcriptase (Gibco-BRL, Grand Island, NY, USA). For PCR, we used DNA polymerase with primers targeting XIAP, Bck-2, c-FLIP (L), Mcl-1 (L), and β-actin. The primer sequences of the targets are listed in Table 1. The amplified products were separated by electrophoresis on a 2% agarose gel and detected under ultraviolet light.

### 4.7. RNA Isolation and Reverse Transcriptase-Quantitative Polymerase Chain Reaction (RT-qPCR)

Total cellular RNA was extracted using TRIzol reagent (Molecular Research Center, Inc., Cincinnati, OH, USA). RNA was quantified using Nanodrop 1000 (Thermo Fisher Scientific, Waltham, MA, USA). Each cDNA was synthesized from 2 μg total RNA using M-MLV reverse transcriptase (Promega, Madison, WI, USA) according to the manufacturer’s instructions. By using the specific primer and SYBR GREEN Premix (TOYOBO, Osaka, Japan), qPCR was performed on the LightCycler^®^ 480 real-time PCR system (Roche Diagnostics, Mannheim, Germany). The primer sequences of XIAP, Bcl-2, c-FLIP (L), Mcl-1 (L), and β-actin are listed in Table 1. The experiments were performed three times, and the expression of each target mRNA was normalized to the housekeeping gene β-actin.

### 4.8. Establishment of Stable Cell Lines Overexpressing Bcl-2, Mcl-1 (L), c-FLIP (L), and Constitutively Active Akt Constructs

A mammalian expression pMAX vector containing the human Bcl-2 gene (provided by Dr Rakesh Srivastava, NIH/NIA) were transfected to Caki cells in a stable manner using Lipofectamine, according to the manufacturer’s instructions (Invitrogen, Carlsbad, CA, USA). The cDNA for Mcl-1 (L) was amplified by PCR with specific primers. The sequences of the sense and antisense primer for Mcl-1 (L) were 5′-GCGACTGGCAAAGCTTGGCCTAA-3′ and 5′-CAACTCTAGAAACTGGTTTTGGTG-3′, respectively. The PCR product was digested with HindIII and XbaI and cloned into the pFLAG-CMV-4TM expression vector (Sigma, St. Louis, MO, USA) and named as pFLAG-CMV-4/Mcl-1 (L). Caki cells were transfected in a stable manner with the pFLAGCMV-4/Mcl-1 (L) plasmid using Lipofectamine according to the manufacturer’s instructions (Invitrogen, Carlsbad, CA, USA). The human cDNA encoding c-FLIP (L) was PCR-amplified from plasmids (pCA-FLAGhFLIP (L), kindly provided by Dr S.I. Park, Korea Centers for Disease Control and Prevention, Seoul, Korea) containing these sequences with the specific primers. The c-FLIP (L) cDNA fragment was digested with KpnI and XhoI and subcloned into the pcDNA 3.1(+) vector (Invitrogen, Carlsbad, CA, USA). The resulting constructs were confirmed by nucleotide sequencing. The Caki cells were transfected in a stable manner with the pCA-FLAGhFLIP (L) plasmid using Lipofectamine, according to the manufacturer’s instructions (Invitrogen, Carlsbad, CA, USA). Akt-overexpressing Caki cells were generated using a vector expressing Myc-His-tagged mouse Aktl (activated) under control of the cytomegalovirus promoter (Upstate Biotechnology, Lake Placid, NY, USA). Caki cells were transfected in a stable manner with the pcDNA3.1-constitutively active Akt and control pcDNA3.1 vector using Lipofectamine, according to the manufacturer’s instructions (Invitrogen, Carlsbad, CA, USA). After 48 h of incubation, transfected cells were selected in cell culture medium containing 700 μg/mL G418 (Invitrogen, Carlsbad, CA, USA). After 2–3 weeks, single independent clones were randomly isolated, and each individual clone was plated separately. After clonal expansion, cells from each independent clone were tested for the expression of Bcl-2, Mcl-1 (L), c-FLIP (L), and Akt proteins by immunoblotting, and the cells were used for further experiments.

### 4.9. siRNA

The XIAP, p53, and PUMA siRNA duplexes were obtained from Cell Signaling Technology (Danvers, MA, USA). Control siRNA duplexes used in this study were purchased from Invitrogen (Carlsbad, CA, USA) and had the following sequence: green fluorescent protein, AAG ACC CGC GCC GAG GUG AAG. Cells were transfected with siRNA oligonucleotides using Lipofectamine RNAiMAX (Invitrogen, Carlsbad, CA, USA) according to the manufacturer’s instructions.

### 4.10. Recording of LVP in Rats

Four male Sprague-Dawley (Orientbio, Inc., Seoul, Korea) rats (200–250 g) were injected intravenously with 1000 IU kg^−1^ heparin and then anaesthetized by intravenous injection of 50 mg kg^−1^ pentobarbital. After they were sedated and had lost pedal reflex activity, the hearts were rapidly excised and perfused via the aorta on a Langendorff apparatus using modified Krebs–Henseleit buffer saturated with carbogen (95% O_2_ and 5% CO_2_) containing 112 mM NaCl, 5 mM KCl, 11.5 mM glucose, 25 mM NaHCO_3_, 1.2 mM MgSO_4_, 1.2 mM KH2PO_4_, 2 mM pyruvic acid, and 1.25 mM CaCl_2_ at a constant perfusion pressure (75 mmHg). An isotonic sodium chloride-filled latex balloon attached to a metal cannula was placed in the left ventricle through the pulmonary vein and connected to a pressure transducer (CyberAmp 380, Axon Instruments, San Jose, CA, USA) for measurement of left ventricular pressure. The hearts were allowed to equilibrate for 10–15 min. All parameters were measured or calculated during 15 min before and after the reperfusion of BMI-1026 with buffer. BMI-1026 was perfused serially from low concentration (1 nM) to high concentration (100 nM). The procedures used in this experiment were reviewed and approved by the Institutional Animal Care and Use Committee at Korea Institute of Toxicology (RS15010).

### 4.11. Clonogenic Assay

Caki cells were seeded into 6-well plates at 1500 cells/well in a final volume of 2 mL medium containing appropriate drug concentrations. Triplicate cultures were used, and the time was tested for each drug concentration. At the end of the drug exposure period, the drug-containing medium was replaced with fresh media. All cultures were incubated for an additional 7 days. The medium was then aspirated, and the dishes were washed once with PBS, fixed with 100% methanol for 30 min, and stained with a filtered solution of 0.5% (*w*/*v*) crystal violet (Sigma, St. Louis, MO, USA) for 30 min. The wells were washed with PBS and dried at room temperature. The colonies, defined as groups of ≥50 cells, were scored manually with an inverted microscope.

### 4.12. Cell Imaging System

Caki cells were cultured in DMEM (Welgene Inc., Gyeongsan, Korea) supplemented with 10% exosome-free FBS (Welgene Inc., Gyeongsan, Korea) in a 96-well culture plate (Corning Life Science, Corning, NY, USA) for 18–24 h according to the manufacturer’s instructions. Using this system, a cell proliferation assay and cell migration assay were performed. Cell proliferation was evaluated (time-curve) by measuring the cell confluence using a real-time imagining system Incucyte^®^ (Incucyte^®^ live-cell ESSEN BioScience, Inc., Ann Arbor, MI, USA). The results of the cell migration assay were assessed using a scratch assay. A scratch was made on confluent monolayers using a 96-pin WoundMaker™ (BioScience, Inc.). Wound images were automatically acquired and registered by the Incucyte^®^ software system. CellPlayer™ 96-Well Invasion Assay software was used to fully automate data collection. Data were processed and analyzed using the Incucyte^®^ 96-Well Cell Invasion Software Application Module.

### 4.13. Statistical Analysis

All data were analyzed by one-way analysis of variance (one-way ANOVA) followed by post hoc comparisons (Student–Newman–Keuls) using SPSS v. 25.0 software (SPSS, Inc., Chicago, IL, USA).

## Figures and Tables

**Figure 1 ijms-22-04268-f001:**
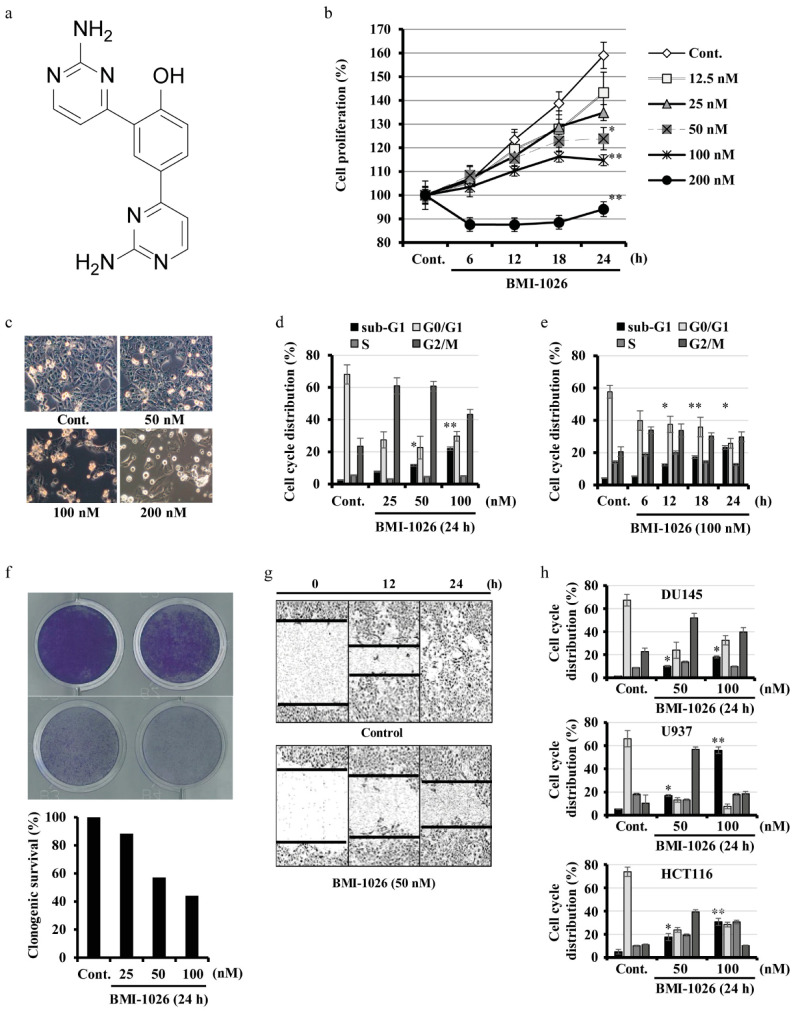
Anticancer effects of BMI-1026 on various human cancer cells. (**a**) Structure of BMI-1026. (**b**) Caki cells were treated with increasing concentrations of BMI-1026 for the indicated time periods, and cell proliferation was then determined with an Incucyte^®^. (**c**,**d**,**f**,**h**) Caki cells were treated with BMI-1026 for 24 h. Morphological changes were visualized by light microscopy (200× magnification) (**c**). Clonogenic survival assay. Cell growth without BMI-1026 was considered as 100% (**f**). (**e**) Caki cells were treated with 100 nM BMI-1026 for the indicated time periods. (**f**) Indicated cells were treated with 50 and 100 nM BMI-1026 for 24 h. Cell cycle distribution was measured using flow cytometry (**d**,**e**,**h**). (**g**) Cell migration was evaluated with an Incucyte^®^. Values in the graphs (**b**,**d**,**e**,**h**) represent the mean ± standard deviation (SD) of three independent experiments. * *p* < 0.05 and ** *p* < 0.01 compared to the control. Cont., control.

**Figure 2 ijms-22-04268-f002:**
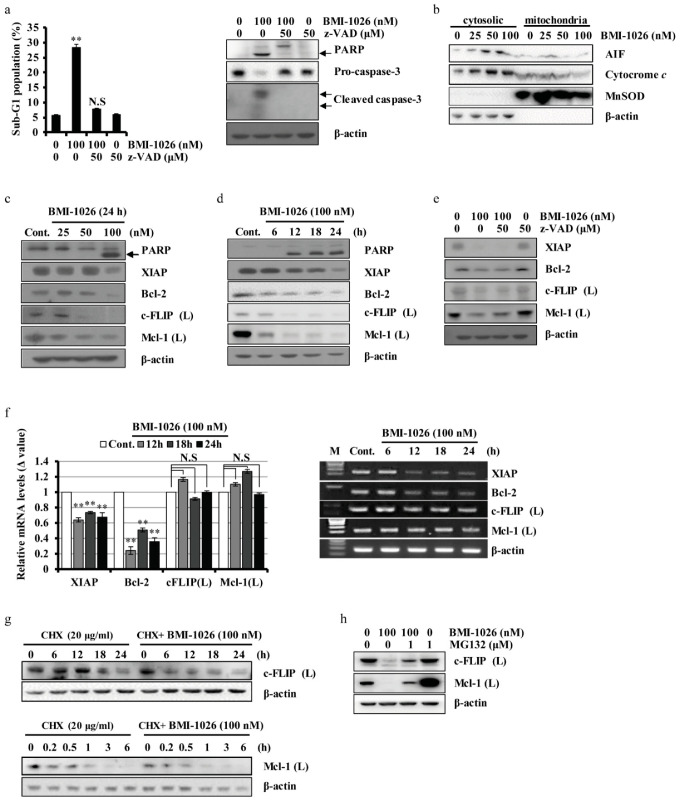
Regulation of apoptosis-related proteins in BMI-1026-treated Caki cells. (**a**,**e**) Caki cells were treated with z-VAD-fmk for 30 min, followed by addition of 100 nM BMI-1026 for 24 h. (**b**,**c**) Caki cells were treated with the indicated concentrations of BMI-1026 for 24 h. (**d**,**f**) Caki cells were treated with 100 nM BMI-1026 for the indicated time periods. (**g**) Caki cells were treated with 20 μg/mL cycloheximide (CHX) in the presence or absence of 100 nM BMI-1026 for the indicated time periods. (**h**) Caki cells were treated with 100 nM BMI-1026 in the presence or absence of 1 mM MG132 for 24 h. Apoptosis and protein expression were analyzed by flow cytometry (**a**) and Western blotting analysis (**a**–**e**,**g**,**h**), respectively. Proteolytic PARP cleavage is indicated by an arrow (**a**,**c**,**d**). Caspase-3 cleavage is indicated by arrows (**a**). The expression level of β-actin was used as a protein loading control (**a**–**e**,**g**,**h**). The expression level of MnSOD was used as a mitochondrial loading control (**b**). mRNA levels were measured using real-time PCR (normalized to the corresponding β-actin mRNAs) and reverse transcription PCR (**f**). Values in the graphs (**a**,**f**) represent the mean ± SD of three independent experiments. ** *p* < 0.01 compared to the control. Cont., control; N.S, not significant.

**Figure 3 ijms-22-04268-f003:**
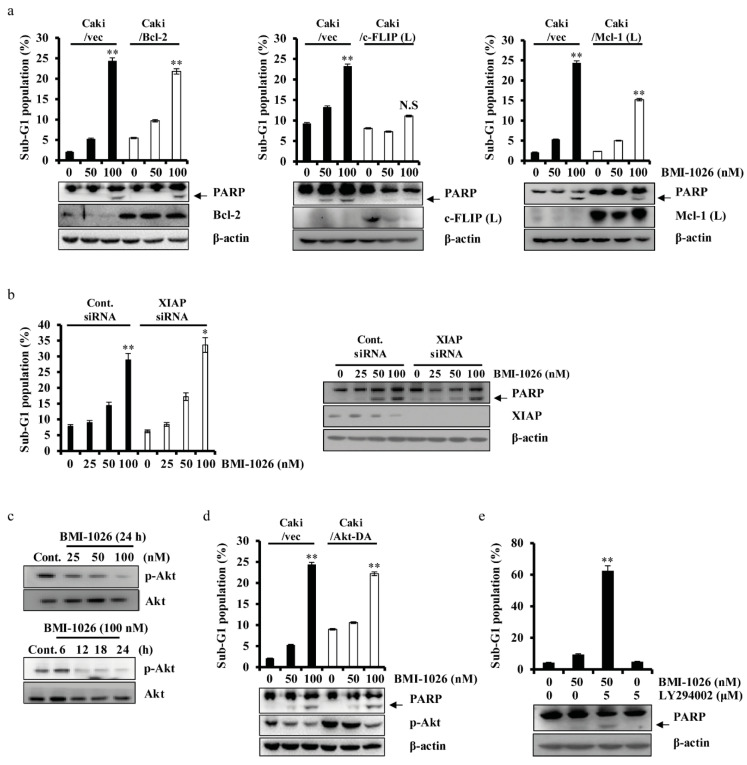
Roles of apoptosis-related proteins in BMI-1026-induced apoptosis. (**a**) Vector cells (Caki/vec), Bcl-2-overexpressing cells (Caki/Bcl-2), c-FLIP (L)-overexpressing cells (Caki/c-FLIP (L)), and Mcl-1 (L)-overexpressing cells (Caki/Mcl-1 (L)) were treated with 50 and 100 nM BMI-1026 for 24 h. (**b**) Caki cells were transfected with control (Cont. siRNA) or XIAP siRNA for 18 h, and the cells were then incubated for another 24 h (0 nM BMI-1026) or treated with the indicated concentrations of BMI-1026 for 24 h. (**c**) Caki cells were treated with increasing concentrations of BMI-1026 for the indicated time periods. (**d**) Caki/vec and the constitutively active form of Akt cells (Caki/Akt-DA) were treated with 50 and 100 nM BMI-1026 for 24 h. (**e**) Caki cells were treated with 50 nM BMI-1026 in the presence or absence of 5 nM LY294002 for 24 h. Apoptosis and protein expression were analyzed by flow cytometry (**a**,**b**,**d**,**e**) and Western blotting analysis (**a**–**e**), respectively. Proteolytic PARP cleavage is indicated with an arrow. The expression level of β-actin was used as a protein loading control. Values in the graphs (**a**,**b**,**d**,**e**) represent the mean ± SD of three independent experiments. * *p* < 0.05 and ** *p* < 0.01 compared to the control. Cont., control; N.S, not significant.

**Figure 4 ijms-22-04268-f004:**
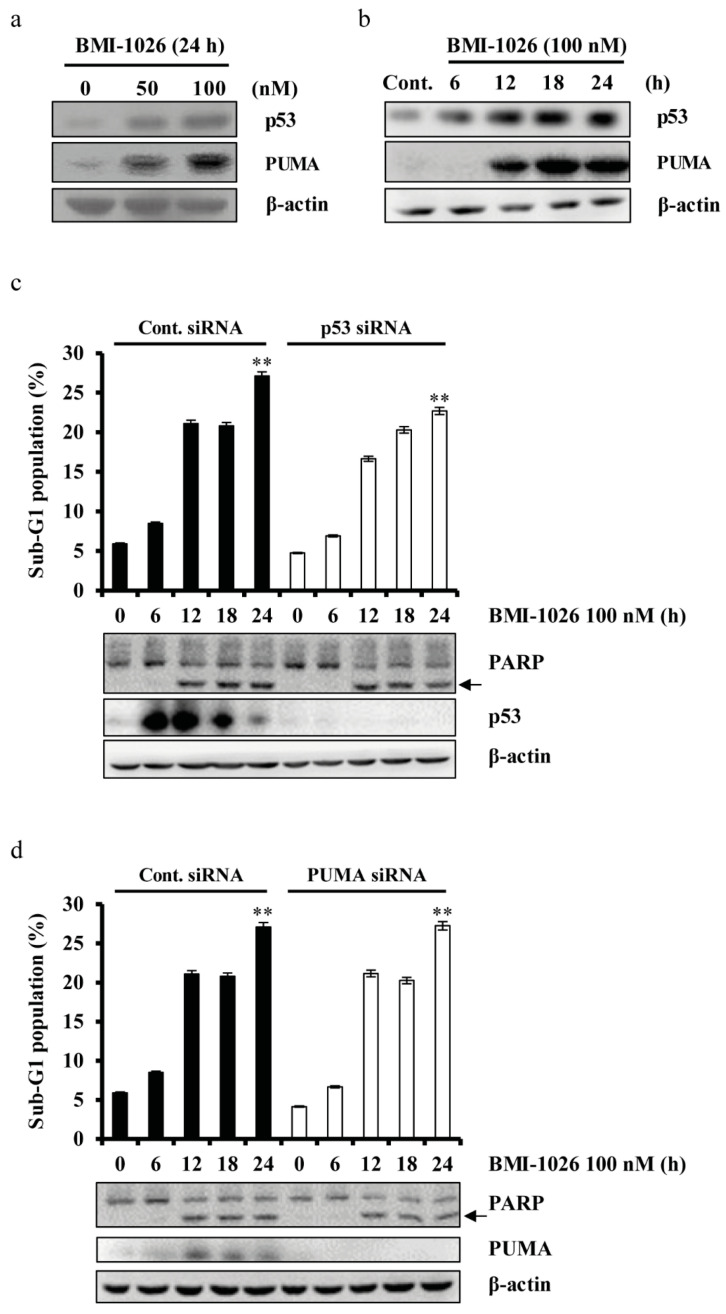
Role of p53 and PUMA in BMI-1026-induced apoptosis. (**a**) Caki cells were treated with the indicated concentrations of BMI-1026 for 24 h. (**b**) Caki cells were treated with 100 nM BMI-1026 for the indicated time periods. (**c**,**d**) Caki cells were transfected with control (Cont. siRNA) or p53 siRNA and PUMA siRNA for 18 h, followed by treatment with 100 nM BMI-1026 for the indicated time periods. Apoptosis and protein expression were analyzed by flow cytometry (**c**,**d**) and Western blotting analysis (**a**–**c**), respectively. Proteolytic PARP cleavage is indicated with an arrow. The expression level of β-actin was used as a protein loading control. Values in the graphs (**c**,**d**) represent the mean ± SD of three independent experiments. ** *p* < 0.01 compared to the control. Cont., control.

**Figure 5 ijms-22-04268-f005:**
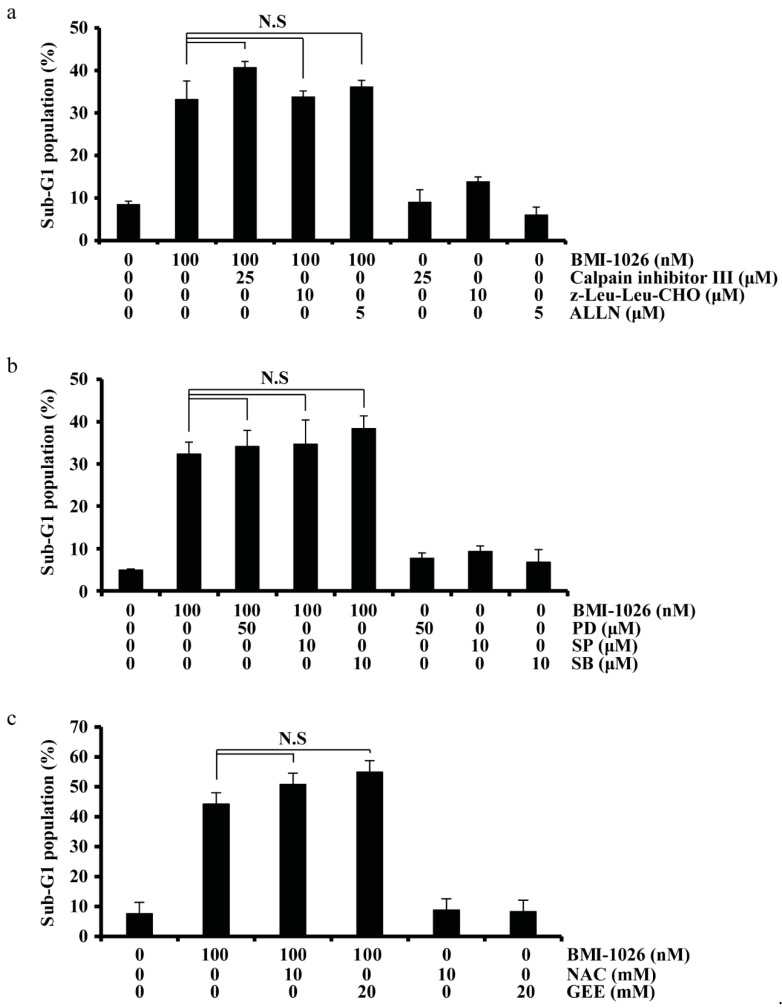
Involvement of various apoptosis-related pathways, including the calpain, mitogen-activated protein kinase, and reactive oxygen species signaling pathways. (**a**–**c**) Caki cells were treated with the indicated drugs for 30 min, followed by addition of 100 nM BMI-1026 for 24 h. Apoptosis was analyzed by flow cytometry (**a**–**c**). Values in the graphs (**a**–**c**) represent the mean ± SD of three independent experiments. N.S, no significance.

**Figure 6 ijms-22-04268-f006:**
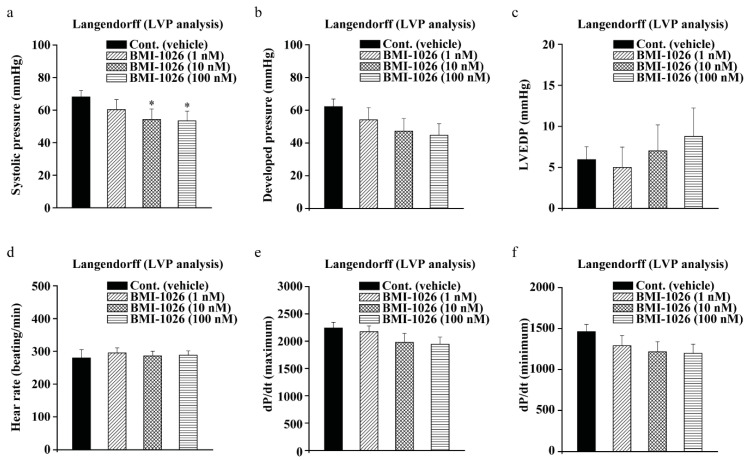
Assessment of electrophysiological safety of BMI-1026 in rats by left ventricular pressure (LVP) analysis. (**a**–**f**) All experiments were measured using rat LVP analysis. (**a**) Effect of BMI-1026 on left ventricular end-systolic pressure. (**b**) Effect of BMI-1026 on left ventricular pressure development. (**c**) Effect of BMI-1026 on left ventricular end-diastolic pressure. (**d**) Effect of BMI-1026 on heart rate. (**e**–**f**) Effect of BMI-1026 on ventricular contractility force. Significant differences were assessed by unpaired *t*-test at * *p* < 0.05. Cont., control; LVEDP, left ventricular end-diastolic pressure.

**Table 1 ijms-22-04268-t001:** Primer sequences.

Gene Name	Sequences *
* XIAP sense *	5′-GCTTGCAAGAGCTGGATTTT-3′
* XIAP antisense *	5′-TGGCTTCCAATCCGTGAG-3′
* Bcl-2 sense *	5′-CAGGTATGCACCCAGAGTGA-3′
* Bcl-2 antisense *	5′-GTCTCT GAAGACGCTGCTCA-3′
* c-FLIP (L) sense *	5′-TGCTGAAGTCATCCATCAGG-3′
* c-FLIP (L) antisense *	5′-ATTCCTAGGGGC TTGCTCT-3′
* Mcl-1 (L) sense *	5′-GCGACTGGCAAAGCTTGGCCTCAA-3′
* Mcl-1 (L) antisense *	5′-CAACTCTAGAAACTGGTTTTGGTG-3′
*β-actin sense*	5′-AATCTGGCACCACACCTTCTA-3′
*β-actin antisense*	5′-ATAGCACAGCCTGGATAGCAA-3′

* Primer sequences of apoptosis-related genes used in both PCR and quantitative PCR (qPCR).

## Data Availability

The data used in the current study are available from the corresponding authors upon adequate request.

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
