# Peer review of "Cyclin-Dependent Kinase Inhibitor BMI-1026 Induces Apoptosis by Downregulating Mcl-1 (L) and c-FLIP (L) and Inactivating p-Akt in Human Renal Carcinoma Cells"

_ijms, 2021, doi:10.3390/ijms22084268_

Round 1

Reviewer 1 Report

BMI-1026, a synthetic 2-aminopyrimidine analog, is a potent CDK inhibitor. Recent studies showed that BMI-1026 has potential anticancer activity. Also, BMI-1026 can inhibit epithelial cell transforming gene 2 phosphorylation in the G2/M phase. However, the effects of BMI-1026 on apoptosis-related genes and related mechanisms are still unclear.

The study by Kim et al aimed to explore the anticancer mechanisms of BMI-1026 on human renal carcinoma Caki cells. First, they confirmed the antiproliferative effects of BMI-1026 on various human cancer cells, including Caki cells. They demonstrated that BMI-1026-induced apoptosis is dependent on caspase-3 activation, the release of AIF and cytochrome c, and Mcl-1 (L) and c-FLIP (L) downregulation. Moreover, the authors showed the electrophysiological safety of BMI-1026 in vitro and in vivo, which is very promising.

This study is very interesting and has high pathophysiological relevance as there is an urgent need for the treatment of cancers. This study will thus offer new insights on improving the treatment of cancers. The findings in this manuscript were innovative and will have an impact on the field.

Overall, this study was well-designed, and the data was clearly presented. One minor issue is please specify how you administrated BMI-1026 to rats in Materials and Methods.

Author Response

Dear reviewer,

We hope you are doing well.

We sincerely appreciate the time and effort of you.

Having received the kind comments by reviewer, we revised our manuscript with attention to each of the comments by reviewer. We appreciate the reviewer very much, who raised the very important critiques to strengthen the claim of our manuscript. We have given very careful consideration to the suggestions and have revised our manuscript. We incorporated new information in the revised version of our manuscript.

     We have fully described the manuscript addressing to the particular issue raised by reviewer. We have responded all the comments by the reviewer point-by-point as follows. We marked light blue on the corrected or added parts on the manuscript in comparison with un-revised manuscript.

     Thank you very much in advance for your time and effort involved in arranging review process.

Yours Sincerely,

Shin Kim, M. D., Ph. D.

The corresponding author

Department of Immunology

Keimyung University School of Medicine

1095, Dalgubeol-daero, Dalseo-Gu, Daegu, 42601, Republic of Korea

Tel: 82-53-258-7359, Fax: 82-53-258-7355

Reviewer’s comments

#1 reviewer :

BMI-1026, a synthetic 2-aminopyrimidine analog, is a potent CDK inhibitor. Recent studies showed that BMI-1026 has potential anticancer activity. Also, BMI-1026 can inhibit epithelial cell transforming gene 2 phosphorylation in the G2/M phase. However, the effects of BMI-1026 on apoptosis-related genes and related mechanisms are still unclear.

The study by Kim et al aimed to explore the anticancer mechanisms of BMI-1026 on human renal carcinoma Caki cells. First, they confirmed the antiproliferative effects of BMI-1026 on various human cancer cells, including Caki cells. They demonstrated that BMI-1026-induced apoptosis is dependent on caspase-3 activation, the release of AIF and cytochrome c, and Mcl-1 (L) and c-FLIP (L) downregulation. Moreover, the authors showed the electrophysiological safety of BMI-1026 in vitro and in vivo, which is very promising.

This study is very interesting and has high pathophysiological relevance as there is an urgent need for the treatment of cancers. This study will thus offer new insights on improving the treatment of cancers. The findings in this manuscript were innovative and will have an impact on the field.

Overall, this study was well-designed, and the data was clearly presented. One minor issue is please specify how you administrated BMI-1026 to rats in Materials and Methods.

Answer: Thank you for your kind advice. As you mentioned, we carefully checked the Materials and Methods. Then we found that there is a critical problem in Materials and Methods. In the original manuscript, it was described that in vivo study was performed, but actually we conducted the Langendorff-perfused rat heart. Therefore, we corrected the description of methodology to specify how we administrated BMI-1026 to rats in Materials and Methods. You can see the corrected description in the revised manuscript (16 page, 440 line). We apologize again for any confusion this may have caused. Please understand our mistakes.

Reviewer 2 Report

The manuscript Cyclin-dependent kinase inhibitor BMI-1026 induces apoptosis by downregulating Mcl-1 (L) and c-FLIP (L) and inactivating p-Akt in human renal carcinoma cells by Kim et al. presents results on the molecular mechanisms of anticancer effects of BMI-1026 in human carcinoma cell line. The authors found that BMI-1026-induced apoptosis of Caki cells is dependent on caspase-3 activation, the release of AIF and cytochrome c, and Mcl-1 (L) and c-FLIP (L) downregulation. Moreover, they also demonstrated the electrophysiological safety of BMI-1026.

In general, the presented data are interesting and propose that BMI-1026 might be a promising anticancer agent. However, the manuscript has several shortcomings that need to be clarified before the acceptance.

Comments and suggestions:

Lane 31: Should read “roles in BMI-1026-induced”

Lane 34: Should read “PI3K/Akt inhibitor LY294002 enhanced Caki cell apoptosis induced by BMI-1026”.

Lane 79: Should read “2.1. BMI-1026 has antiproliferative effects and induces apoptosis in various human cancer cells”.

Lane 87: The authors analysed the sub-G0 cell population by employing the strategy of PI staining/flow cytometry. Authors might provide the results of cell cycle phase changes (G1, S, G2/M) of BMI-1026-treated cells. This will add significant value to Figure 1d and 1e. The findings of changes in cell cycle distribution of Caki cells after BMI-1026 treatment should be discussed within the contents of other data.

Figure 1a: I would suggest the authors to present only the structure of BMI-1026 and specify its final purity in the Methods section. The strategy of synthesis seems to be redundant as yield of intermediates, the purities, and MS/NMR analyses are not described in sufficient detail.

Figure 1e: Authors should check the labelling for statistical significance (there is one asterisks for 24 hrs vs two asterisks for 18 hrs).

Figure 2c and 2d: There is obvious discrepancy between the expression of c-FLIP(L) and Mcl-1(L) when 100 nM BMI-1026 was used for 24 h. In Figure 2c there is no expression detected for c-FLIP(L) and expression of Mcl-1(L) is detected. This contradicts the results presented in Figure 2d (c-FLIP(L) detected and Mcl-1(L) not). Please, explain.

Figure 2g: If possible, the blot of better quality for c-FLIP(L) should be provided.

Authors might consider to support their data on detection of sub-G0 cell fractions (Figure 5) by any more sensitive assays, e.g. analysing the caspase-3 or caspase-9 activation by colorimetric or fluorometric assays/kits.

Lanes 322-323: The U937 cell line should not be mentioned here, as these cells were grown in RPMI medium as is mentioned later (lanes 326-327). Correct it, please.

Author Response

Dear reviewer,

We hope you are doing well.

We sincerely appreciate the time and effort of you.

Having received the kind comments by reviewer, we revised our manuscript with attention to each of the comments by reviewer. We appreciate the reviewer very much, who raised the very important critiques to strengthen the claim of our manuscript. We have given very careful consideration to the suggestions and have revised our manuscript. We incorporated new information in the revised version of our manuscript.

     We have fully described the manuscript addressing to the particular issue raised by reviewer. We have responded all the comments by the reviewer point-by-point as follows. We marked light blue on the corrected or added parts on the manuscript in comparison with un-revised manuscript.

     Thank you very much in advance for your time and effort involved in arranging review process.

Yours Sincerely,

Shin Kim, M. D., Ph. D.

The corresponding author

Department of Immunology

Keimyung University School of Medicine

1095, Dalgubeol-daero, Dalseo-Gu, Daegu, 42601, Republic of Korea

Tel: 82-53-258-7359, Fax: 82-53-258-7355

#2 reviewer :

The manuscript Cyclin-dependent kinase inhibitor BMI-1026 induces apoptosis by downregulating Mcl-1 (L) and c-FLIP (L) and inactivating p-Akt in human renal carcinoma cells by Kim et al. presents results on the molecular mechanisms of anticancer effects of BMI-1026 in human carcinoma cell line. The authors found that BMI-1026-induced apoptosis of Caki cells is dependent on caspase-3 activation, the release of AIF and cytochrome c, and Mcl-1 (L) and c-FLIP (L) downregulation. Moreover, they also demonstrated the electrophysiological safety of BMI-1026.

In general, the presented data are interesting and propose that BMI-1026 might be a promising anticancer agent. However, the manuscript has several shortcomings that need to be clarified before the acceptance.

Comments and suggestions:

Lane 31: Should read “roles in BMI-1026-induced”

Answer: Thank you for your kind advice. As you mentioned, we corrected the sentence. You can see the corrected description in the revised manuscript (1 page, 31 line).

Lane 34: Should read “PI3K/Akt inhibitor LY294002 enhanced Caki cell apoptosis induced by BMI-1026”.

Answer: Thank you for your kind advice. As you mentioned, we corrected the sentence. You can see the corrected description in the revised manuscript (1 page, 34 line).

Lane 79: Should read “2.1. BMI-1026 has antiproliferative effects and induces apoptosis in various human cancer cells”.

Answer: Thank you for your kind advice. As you mentioned, we corrected the sentence. You can see the corrected description in the revised manuscript (2 page, 79 line).

Lane 87: The authors analysed the sub-G0 cell population by employing the strategy of PI staining/flow cytometry. Authors might provide the results of cell cycle phase changes (G1, S, G2/M) of BMI-1026-treated cells. This will add significant value to Figure 1d and 1e. The findings of changes in cell cycle distribution of Caki cells after BMI-1026 treatment should be discussed within the contents of other data.

Answer: Thank you for your kind advice. All authors fully agreed your opinion that ‘This will add significant value to Figure 1d and 1e’. As described in Introduction part, BMI-1026 is a potential anticancer agent that targets cyclin-dependent kinases. However, when we reviewed the literature, there were few studies on the apoptotic mechanisms of BMI-1026. Actually, we wanted to investigate the apoptotic mechanisms rather than the cancer cell cycle arrest mechanisms of BMI-1026 in Caki cells. So, we just revealed the sub-G0 cell population in this study although BMI-1026. However, as mentioned above, we performed the experiment following your advice that the results whether BMI-1026 targeting cyclin-dependent kinases actually induces cell cycle arrest is of importance. As shown in Figure 1d and 1e, we analyzed the cell cycle distribution of BMI-1026-treated Caki cells in dose- and time-dependent manners. Moreover, we performed the analysis of cell cycle distribution of BMI-1026-treated DU145, U937, and HCT116 cells (Figure 1h). Additionally, we described the effect of BMI-1026 in the Discussion part. You can see the corrected description in the revised manuscript (2 page, 86, 87, 88, 94, 95, 97, and 108 lines; 11 page, 256 line).

Figure 1a: I would suggest the authors to present only the structure of BMI-1026 and specify its final purity in the Methods section. The strategy of synthesis seems to be redundant as yield of intermediates, the purities, and MS/NMR analyses are not described in sufficient detail.

Answer: Thank you for your kind advice. As you mentioned, we presented only the structure of BMI-1026 in Figure 1a, and corrected the legend of Figure 1a according to the Figure 1a. Moreover, the strategies of synthesis of intermediates have been removed. We added the purities and MS/NMR analyses according to reviewer’s comments. You can see the corrected description in the revised manuscript (13 page, 330 line).

Figure 1e: Authors should check the labelling for statistical significance (there is one asterisks for 24 hrs vs two asterisks for 18 hrs).

Answer: Thank you for your kind advice. As you mentioned, we checked the statistical significance. We performed the one-way ANOVA followed by post-hoc comparisons (Student-Newman-Keuls) using SPSS software. As a result of statistical analysis, 18 hrs showed statistical significance at p value < 0.01 compared to the control. Moreover, 24 hrs showed statistical significance at p value < 0.05 compared to the control. Therefore, we described the meanings of those in the figure legend of Figure 1.

Figure 2c and 2d: There is obvious discrepancy between the expression of c-FLIP(L) and Mcl-1(L) when 100 nM BMI-1026 was used for 24 h. In Figure 2c there is no expression detected for c-FLIP(L) and expression of Mcl-1(L) is detected. This contradicts the results presented in Figure 2d (c-FLIP(L) detected and Mcl-1(L) not). Please, explain.

Answer: Thank you for your kind advice. As you mentioned, we carefully checked the result in Figure 2c and 2d. We found that this discrepancy occurred because the development time was different when the Western blotting band was detected. Actually, we would like to show that BMI-1026 can down-regulate apoptosis-related proteins and tends to reduce apoptosis-related proteins. However, we performed the time kinetics again in consideration of the reviewer’s comment. Then we corrected the expression results of c-FLIP (L) and Mcl-1 (L) in Figure 2d. You can see the corrected blots in the revised manuscript (5 page).

Figure 2g: If possible, the blot of better quality for c-FLIP(L) should be provided.

Answer: Thank you for your kind advice. As you mentioned, we performed immunoblot for better quality of c-FLIP (L) in Figure 2g. You can see the improved quality of the blot in the revised manuscript (5 page).

Authors might consider to support their data on detection of sub-G0 cell fractions (Figure 5) by any more sensitive assays, e.g. analysing the caspase-3 or caspase-9 activation by colorimetric or fluorometric assays/kits.

Answer: Thank you for your kind advice. As you mentioned, we carefully checked the result in Figure 5. Actually, we wanted to identify the involvement in BMI-1026-induced apoptosis in Caki cells. However, none of the apoptosis-related signaling pathway inhibitors affected BMI-1026-induced apoptosis in Figure 5. Of course, caspase-3 is a major marker of apoptosis, and changes in a component of apoptosis signaling pathways such as caspase-3 did not affect the apoptosis sub-G0 population. Therefore, we concluded that the apoptosis-related signaling pathways were not associated with BMI-1026-induced apoptosis.

Lanes 322-323: The U937 cell line should not be mentioned here, as these cells were grown in RPMI medium as is mentioned later (lanes 326-327). Correct it, please.

Answer: Thank you for your kind advice. As you mentioned, we corrected the sentence. You can see the corrected description in the revised manuscript (13 page, 325 line).

Round 2

Reviewer 2 Report

Authors have addressed my concerns adequately.

Minor comment:

Fig. 1h: The label for subG0, G0/G1, S and M phases should be added for DU145, U937 and HCT116 cells.